



# Rainfall-runoff processes in the Loess Plateau, China: Temporal dynamics of event rainfall–runoff characteristics and diagnostic analysis of runoff generation patterns

Qiang Wu[1,2], Zhaoxi Zhang[3], Guodong Zhang[3], Shengqi Jian[1,2], Li Zhang[1,2], Guang Ran[1,2], Dong Zhao[1,2], Xizhi Lv[4], and Caihong Hu[1,2]

[1]School of Water Conservancy & Engineering, Zhengzhou University, University, Zhengzhou, 450001, China
[2]Yellow River Institute for Ecological Protection & Regional Coordinated Development, Zhengzhou, 450001, China
[3]Henan Yellow River Hydrological Survey and Design Institute, Zhengzhou, 450001, China
[4]Key Laboratory of Soil and Water Loss Process and Control on the Loess Plateau of Ministry of Water Resources, Yellow River Institute of Hydraulic Research, Zhengzhou, 450001, China

**Correspondence:** Caihong Hu (hucaihong@zzu.edu.cn)

**Abstract.** The Loess Plateau is the most erosion-prone area in China, while under large-scale ecological restoration runoff and sediments continue to decrease. This study examined the runoff generation mechanism at the catchment scale to understand the change in runoff generation. Six base flow separation method were tested and the nonparametric simple smoothing method was used to seperating base flow. With the event runoff separation procedure, 340 rainfall–runoff events are selected in five typical

catchments affected by significant human intervention in the Loess Plateau. Runoff characteristics, such as the event runoff coefficient, time scale, rise time, and peak discharge are studied on monthly and long-term scales. In catchments of Jialuhe, Chabagou and Gushanchuan with poor vegetation runoff response is strongly decided by rainfall intensity and is produced by Horton overland flow (HOF). While the mountainous catchments of Jingle and Zulihe runoff response is controlled by rainfall volume. The relation between runoff event characteristics and rainfall is complicated in Loess Plateau, where rainfall

and underlying surface is significantly changing. The monthly of event characteristics is mostly controlled by rainfall characteristics. Long-term runoff coefficient experiences decreasing trend, while time scale trend is increasing. Land use changes lead to increasing catchment wetness display mostly strong reason in event characteristic response. According to our proposed framework for classifying dominant runoff generation patterns considering of hydrograph response time, discharge source, and flow paths, HOF runoff is still the dominant mechanism, but gradually shifts to Dunne overland flow (DOF) and combination

runoff. We speculate that the reduction in runoff in the Yellow River is likely to be the dominant runoff mechanism changing.

## 1   Introduction

The Loess Plateau is the deepest and largest loess deposit in China and is prone to erosion owing to episodes of short and heavy precipitation (Shi and Shao, 2000). Many soil and water conservation practices (e.g., terracing, check-dam construction, natural vegetation rehabilitation, and planting of vegetation) have been implemented since the 1970s (Fu et al., 2017; Wu

et al., 2019; Zhao et al., 2013) with economic and societal development in China. Waterflow and sediment have decreased





significantly owing to large-scale restoration practices and climate change (Zuo et al., 2016), and runoff generation from the Loess Plateau has decreased since the 1970s (Jiongxin, 2005). From 1990–2000, annual recharge runoff to the sea was reduced to $13.2 \times 10^9$ m$^3$ year$^{-1}$, 28.7% that in the 1950s ($48.0 \times 10^9$ m$^3$ year$^{-1}$) (Yu et al., 2011). The generation of storm-flow and the factors affecting hydrological processes owing to the changing environment are central hydrological research themes in

the Loess Plateau. The Yellow River flows through an important economic area, which accounts for 26.5% of the GDP of China. Water resource management and flood forecasting affect 420 million people. The relationship between rainfall and runoff is relevant to many practical problems in the Yellow River Basin (Xi, 2019), and the generation of runoff and associated processes are important for understanding flood generation and sediment mobilization (Zhang et al., 2018). The Loess Plateau is located in arid and semi-arid areas where hydrological processes are more variable with the changing climate and the influence

of anthropogenic activities (Gao et al., 2016). Additionally, the influence of land use and cover change (LUCC) directly or indirectly influences cloud cover, albedo, evapotranspiration, soil moisture, vegetation water use, and streamflow (Pielke Sr et al., 2011). Climate change affects the inputs of hydrological processes, such as precipitation, temperature, wind, radiation, and evaporation (Stagl et al., 2014), which modify regional runoff generation mechanisms. Thus, studying runoff generation is imperative for understanding water and sediment changes in the Yellow River.

Techniques used for learning about runoff generation are based on (1) runoff isotopic or chemical characteristics (Tetzlaff et al., 2009), (2) hydroclimatic characteristics (Dunne, 1978; Mirus and Loague, 2013), or (3) hydrograph characteristics (Hannah et al., 2000; Sawicz et al., 2011; Toth, 2013). Costly isotopic and chemical methods are only used in small field experiments (Fan et al., 2016; Pan et al., 2018; Gou et al., 2019). Runoff generation at the catchment scale needs to be understood for large-scale ecological restoration. Thus, hydrographs and environmental characteristics are used to identify dominant runoff

mechanisms. Continuous streamflow is considered as a series of rainfall-runoff events that add base flow. This thought was proposed by Seibert et al. (2016) for analyzing large-scale changes in hydrological processes and runoff generation mechanisms. Catchment runoff production processes and the temporal dynamics of runoff generation processes can be studied by associating runoff hydrographs and catchment wetness (Graeff et al., 2012; Blume et al., 2007). Large-scale ecological restoration in the Loess Plateau changes runoff mechanisms and sediment generation by affecting the response of runoff to rainfall in different

regions (James and Roulet, 2007; Stieglitz et al., 2003). Moreover, climate and land use changes reshape the relationships between hydrological processes and changes the response of basins to rainfall events (Merz et al., 2012). Thus, discovering and analyzing the variability of an event is important for water resource management and flood forecasting and help explain how runoff generation mechanisms in the Loess Plateau experience long-term changes.

Runoff events are the elementary unit used to study hydrological processes. However, the methods for identifying these

events are not widely accepted. Existing methods involve three steps: (1) separating the base flow, (2) finding starting and ending points, and (3) calculating rainfall for runoff events. However, these applied methods and presumptions are often only suitable for specific basins or regions. For example, event-based recession analysis (Blume et al., 2007), recursive filters (Chapman et al., 1996; Eckhardt, 2005), and simple time series smoothing (IH, 1980) have been applied for base flow separation, but the lack of *true* base flow (Su et al., 2016) creates the need for additional thresholds that vary by region (Hewlett and

Hibbewitrt, 1967). The Loess Plateau experiences heavy rainfall events during short periods and distinct runoff changes. Thus,



the beginning of runoff can be easily determined, but the end point is difficult to identify, and additional separation thresholds, such as the long-term base flow index or quick and base flow ratio, are needed (Mei and Anagnostou, 2015). However, few researches studied how to identify a runoff event in the Loess Plateau.

Runoff variables, such as the event runoff coefficient and a time scale are used to provide insight of changing long-term event characteristics for a spatio-temporal analysis of hydrological processes (Merz and Blöschl, 2009b, a; Seibert et al., 2016; Gaál et al., 2012). Researchers have studied the relationship of the event runoff coefficient with rainfall intensity (Blume et al., 2007; Graeff et al., 2012), event rainfall volume (Blume et al., 2007; Merz et al., 2006), and runoff and soil moisture (Graeff et al., 2012; James and Roulet, 2007; Merz et al., 2006; Rodríguez-Blanco et al., 2012) to find and highlight dependence. The time scale and runoff coefficient vary with the seasonality of rainfall and soil moisture (Merz and Blöschl, 2009b; Rodríguez-Blanco 65 et al., 2012). However, long-term and large-scale land use changes (e.g., returning farmland to forest, check-dam construction, and terraces) can change these variables (Yang et al., 2019; Li et al., 2017; Ran et al., 2014). Additionally, climate change (e.g., precipitation and temperature) might also affect hydrological processes (Ajami et al., 2017; Dumanski et al., 2015; Sawicz et al., 2014). The spatio-temporal characteristics of hydrological processes help explain runoff mechanisms in catchments.

The runoff process is decided by multiple factors, such as topography, soil properties, rainfall volume and intensity, vegeta- 70 tion, and land use. The variable source concept runoff generation process was first illustrated by Dunne (1978). Figure 1 shows the major controls related to runoff processes. Horton overland flow (HOF) is prone to occur in arid and semi-arid regions, whereas Dunne overland flow (DOF) occurs in areas with longer-duration rainfall and humid soil surfaces. Many scholars have studied the impact of climate and human activity on changing runoff patterns in the sharp water and sediment reduction conditions of the Yellow River Basin (Wu et al., 2018; Feng et al., 2016), but the few studies using event runoff to analyze 75 runoff generation on a catchment scale are mostly concentrated in Switzerland, the United States, and Australia (Vivoni et al., 2008; Votrubova et al., 2017; Zehe et al., 2010). Additionally, few studies investigate the long-term spatio-temporal transition of basin-scale runoff generation mechanisms. Thus, these restrictions do little to illustrate the event runoff response in the Loess Plateau. Moreover, few studies use quantitative analysis to study the effects of land use and climate on the characteristics of runoff events. The studies using event runoff identification techniques cannot be adapted to the Loess Plateau because of 80 geographic specificities. Studying the runoff response of land use and climate change under large-scale ecological construction is important to guide basin environmental protection and water resource management in the Yellow River Basin. Additionally, the dominant runoff generation types in a catchment are complicated and need to be studied in depth.

This study had three objectives: (1) to test different methods to separate base flow and identify rainfall–runoff events in the Yellow River Basin, (2) to analyze the temporal dynamics (i.e., monthly and long-term changes, event variability in each 85 catchment) of runoff characteristics for 340 events in five catchments of the Loess Plateau from to 1960–2014 by comparing dynamics and explanatory variables to provide an insight into the drivers of runoff characteristics at different time resolutions (a comparison paper is considered to address the spatial patterns of more complex runoff generation patterns), and (3) to diagnose runoff generation patterns at the basin scale. The reasons for long-term discharge reduction from the Loess Plateau are studied in terms of catchment runoff generation patterns to make fundamental progress in the prevention of flooding hazard and the 90 management of water resources in the Yellow River.





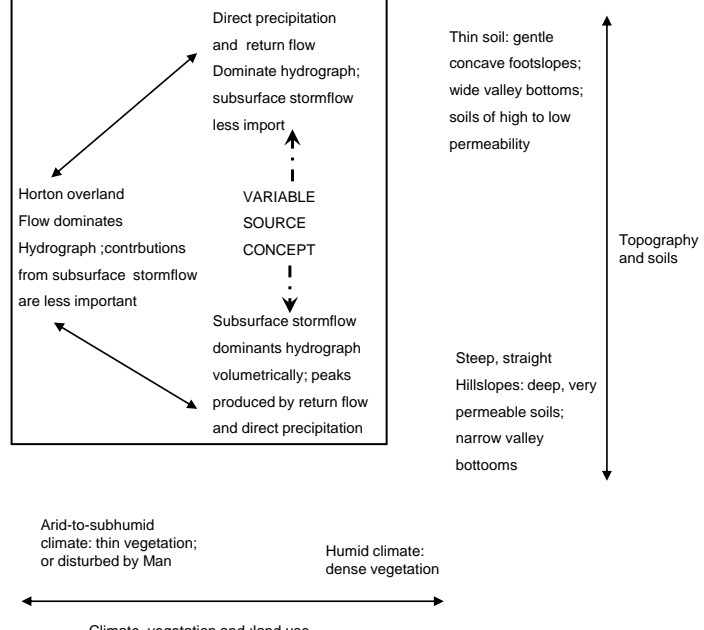

**Figure 1.** Schematic illustration of the occurrence of various runoff processes in relation to their major controls

## 2 Data

The Loess Plateau (100°54′-114°33′E, 33°43′-41°16′N) is located in a transition zone between semi-arid and semi-humid continental climates and has a total area of 624,000 km$^2$. The northwest has a semi-arid continental climate, and the east and south have a semi-humid continental climate. Average precipitation from 1960–2000 ranged from 200–700 mm. Figure
2 shows the principal landscapes of the Loess Plateau and the five sub-catchments selected for this research. Table 1 lists the attributes of each basin.

Water resource management measures have a long history in the Yellow River. Study areas were selected in regions without large reservoirs to avoid the effect of human activity on the discharge events. The five catchments have areas of 187–10,653 km$^2$—Chabagou is the smallest. The Jingle, Jialuhe, and Gushanchuan catchments cover 2799, 1117, and 1263 km$^2$, respec-
tively. The Zulihe basin has the largest area (10,635 km$^2$). The runoff and meteorological data series covers 1965–2014.

Runoff time series were aggregated from the *Hydrographic Yearbooks of the People's Republic of China* published by the Yellow River Conservancy Commission of the China Ministry of Water Resources. The precipitation data at 53 meteorological stations (Figure 2) were obtained from the Yellow River Hydrological Database. Precipitation in the five catchments was estimated by kriging interpolation. Table 2 shows the runoff and precipitation series in the five catchments, for which 340
rainfall-runoff events were selected.



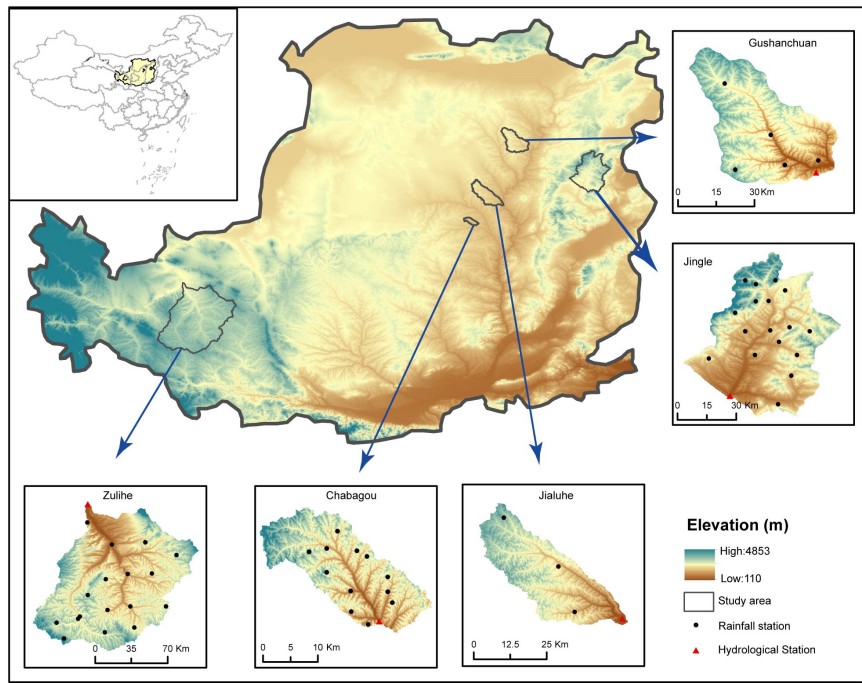

**Figure 2.** Study area of the Loess Plateau, China. Five sub-catchments were selected with significant anthropogenic modifications and LUCC changes. Gushanchuan, Jingle, Jialuhe, and Chabagou are in the primary sediment generation area.

**Table 1.** Principal characteristics of the studied catchments

|  | Zulihe | Chabagou | Jialuhe | Gushanchuan | Jingle |
|---|---|---|---|---|---|
| Catchment area (km$^2$) | 10,653 | 187 | 1,117 | 1,263 | 2,799 |
| X coordinate (UTM) | 35°18′∼36°34′N | 37°47′∼37°37′N | 38°00′∼38°28′N | 39°00′∼39°26′N | 38°14′∼39°00′N |
| Y coordinate (UTM) | 104°12′∼105°33′E | 109°46′∼110°03′E | 109°55′∼110°30′E | 110°31′∼111°02′E | 111°42′∼112°26′E |
| Minimum elevation (m.a.s.l.) | 1388 | 913 | 703 | 796 | 1194 |
| Maximum elevation (m.a.s.l.) | 2820 | 1281 | 1374 | 1402 | 2763 |
| Mean altitude (m) | 1956 | 1072 | 1126 | 1188 | 1716 |
| Aspect | North | South | South | South | South |
| Annual rainfall (mm) | 315 | 416 | 422 | 410 | 450 |



**Table 2.** Rainfall stations, runoff, and precipitation time series in the five catchments

| Basins | Rain stations | Flood events | Time periods |
|---|---|---|---|
| Zulihe | 16 | 15 | 2006—2014 |
| Chabagou | 12 | 51 | 1960—2006 |
| Jialuhe | 3 | 51 | 1960—2006 |
| Gushanchuan | 5 | 125 | 1965—2006 |
| Jingle | 17 | 98 | 1971—2014 |
| Total | 53 | 340 | |

Catchment wetness preceding runoff events was calculated by antecedent impact precipitation (instead of soil moisture), because few rainfall stations observed catchment soil moisture before 2000; the few soil moisture observations constitute point data. Antecedent impact precipitation was calculated using equations (1) and (2). Changes in land use in the research areas were studied by CCI-LC products (CCI website), available from 1992–2015.

$$P_{a,t} = KP_{t-1} + K^2 P_{t-2} + \cdots + K^n \left( P_{a,t-n} + P_{t-n} \right), \tag{1}$$

$$P_{a,t+1} = K \left( P_{a,t} + P_t \right), \tag{2}$$

where $P_{a,t}$ and $P_{a,t+1}$ are the $t$ and $t+1$ day antecedent impact precipitation, respectively (mm). $P_t$ is the rainfall on day $t$ (mm); $P_{t-1}$ and $P_{t-2} \ldots$ are the rainfall (mm) 1 and 2 days, $\ldots$, before $t$; and $K$ is the soil daily reduction factor.

## 3 Methods

### 3.1 Identification of rainfall-runoff events

The three steps used to characterize an event included separating the base flow, identifying runoff events, and attributing rainfall events. Continuous streamflow is divided into base and quick flows by runoff separation into the beginning and end of runoff events, and runoff peaks (Figure 3). The first step is base flow separation. Multiple base flow separation methods were examined to find the most suitable for recognizing runoff events.

The types of base flow separation were tested as follows:

- Single-parameter digital filtering methods by Lyne & Hollick (Lyne and Hollick, 1979), Chapman (Chapman, 1999), and Chapman & Maxwell (Chapman et al., 1996) were tested. The Lyne & Hollick digital filtering technique was first used by in 1979, and introduced into hydrology by Nathan and McMahon in 1990 for basic flow separation. Chapman improved the Lyne-Hollick filtering method in 1991 and proposed the Chapman filtering method. Chapman and Maxwell weight averaged the surface flow and prevailing base flow in 1996 and proposed the Chapman-Maxwell filtering method.



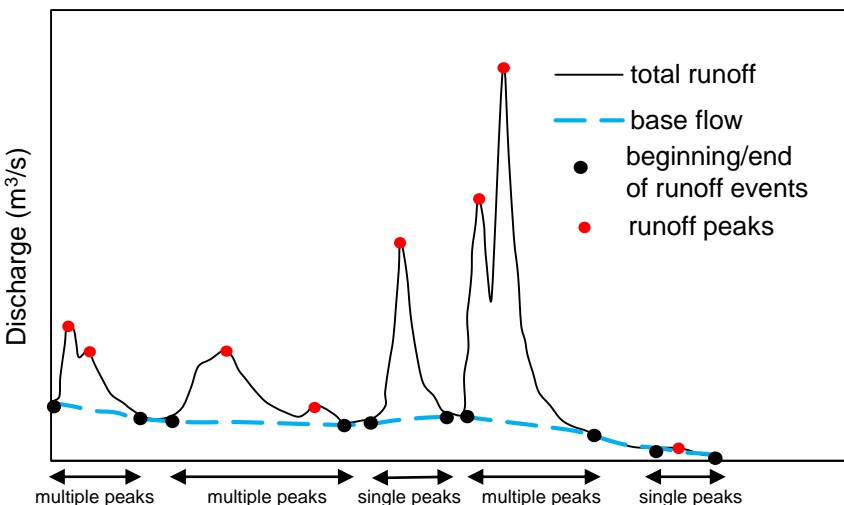

**Figure 3.** Base flow separation and identification of runoff events

– Another recursive digital filtering method proposed by Eckhardt (2005) contains two digital filtering parameters: the base flow index (*BFI*) and a recession constant. The method used to estimate the recession constant was referred to in Vogel and Kroll (1992). *BFI* was computed by the Q90 to Q50 ratio (Smakhtin, 2001).

– A computer program for streamflow hydrograph separation and analysis method (HYSEP), also used in this study, was
first proposed by Petty and Henning in 1979 and primarily used by the US Geological Survey. We chose the P & H fixed window method for base flow separation.

– A nonparametric method was used to analyze runoff time series (IH, 1980). First, the local minimum in the 24-hour window was found in the time series. Next, the turning points were examined by the minimum values of the time series. The turning points were defined as 1.11 times smaller than the nearest minimum value. After identifying turning points,
the method of linear interpolation was used to reconstruct the base flow hydrograph.

The peak, beginning, and end points of event runoff were identified from continuous streamflow by separating the base flow (Figure 3). The beginning and end of rainfall attributed to runoff events were identified by the median basin lag time. This criterion is a physical-based value (Mei and Anagnostou, 2015).

## 3.2 Classification of dominant runoff generation process

The dominant runoff generation process was studied after identifying rainfall–runoff events. We classified the dominant runoff generation process of every event to study the effects of climate change and land use on runoff. Under changing runoff and catchment characteristics, the dominant runoff production patterns varied among events. Every rainfall-runoff event was analyzed with the classification framework, and the results were used for testing the long-term changes and developmen





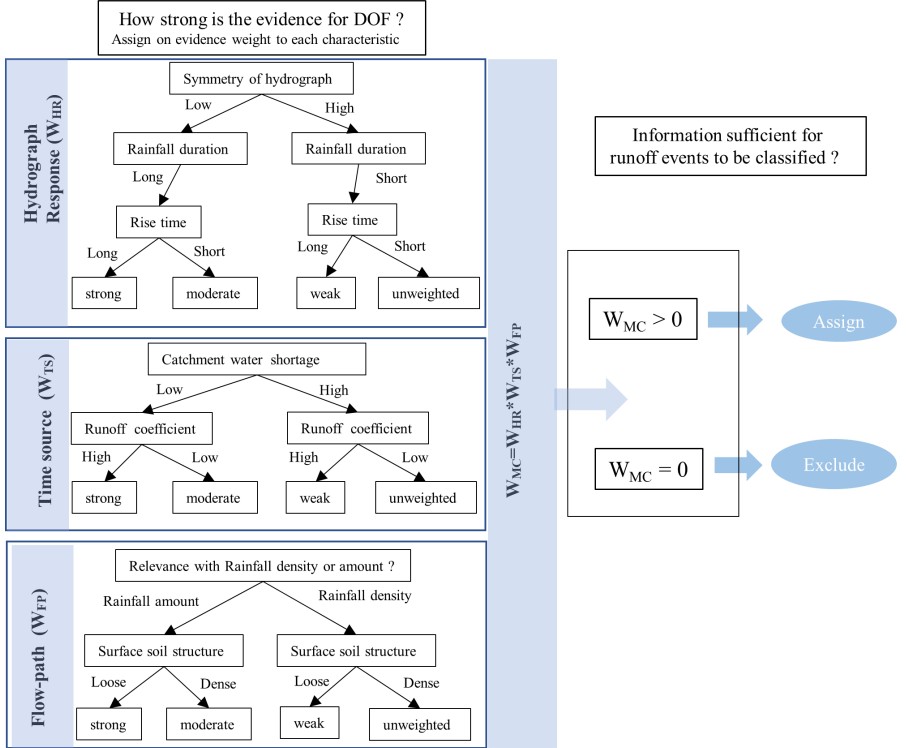

**Figure 4.** DOF evaluation flow chart and classification procedure

The runoff pattern of each event was classified into three classes: HOF (infiltration-excess overland flow), DOF (saturation-excess overland flow), and a combination of HOF and DOF. Three primary runoff event factors (time source, flow path, and hydrograph response) were chosen to classify the runoff process by considering rainfall-runoff characteristics. This classification method was based on the Dunne runoff mechanism diagram (Figure 1). DOF rarely appears in the semi-arid and semi-humid climates of the Loess Plateau. Every event was tested for DOF runoff; Figure 4 shows the DOF identification framework.

Classification evaluation requires a uniform routine with clear rules to fairly compare the results. The event time source ($W_{TS}$), flow-path ($W_{FP}$), and hydrograph response ($W_{HR}$) were used to evaluate all rainfall-runoff events. The hydrograph response was quantified by symmetry, rainfall duration, and rise time. The time source was quantified by catchment water shortage and runoff coefficient. New water fraction contributes more to stormflow with a larger runoff coefficient and smaller catchment water shortage. The relationship of the hydrograph with the rainfall density and amount and with surface soil structure are selected to study the flow path and dominant runoff processes for each event. The degree that DOF affected the rainfall-runoff characteristics was also evaluated by selecting the weight ($W$) to estimate the evidence of relevant characteristics using a numeric rating system.





The evidence weight for the hydrograph response, time source, and flow path represents the support of rainfall-runoff characteristics. Three evidence weights ($W_{TS}$, $W_{FP}$, and $W_{HR}$) were used to generate $W_{MC}$, which tested inferred runoff generation

patterns. The weight of $W_{MC}$ was computed using the following equation:

$$W_{MC} = W_{HR} * W_{TS} * W_{FP}. \tag{3}$$

with $W$ = evidence weight for hydrograph response ($W_{HR}$), time source ($W_{TS}$), and flow path ($W_{FP}$). The final evidence weight $W_{MC}$ ranges from 0–1. A value of 0 denotes the lack of evidence for at least one characteristic. Larger $W_{MC}$ values represent a higher relevance of rainfall–runoff on the runoff generation mechanism. The flow chart in Figure 4 shows the steps

for assigning evidence weights.

We defined four strong, moderate, weak, and unweighted classes to estimate evidence weight. A numeric rating system was created using the following rankings:

$$W_{characteristic} = \left\{ \begin{array}{l} 1 \rightarrow strong, \\ 0.5 \rightarrow moderate, \\ 0.1 \rightarrow weak, \\ 0 \rightarrow unweighted, \end{array} \right\} \tag{4}$$

## 3.3 Temporal dynamics of event rainfall-runoff response

How the temporal dynamics of event rainfall-runoff response were studied by analyzing runoff generation process characteristics. According to identified rainfall-runoff events, multiple temporal scales of monthly and long-term changes were studied using event runoff response dynamics. Characteristic event responses reflect the variability of catchment rainfall-runoff processes. The hypothesized rainfall-runoff drivers were examined to estimate event controls at each catchment.

### 3.3.1 Event characteristics

The runoff coefficient (*rc*), runoff time scale (*ts*), rise time (*rt*), and peak discharge (*peak*) were used to quantitatively compare rainfall-runoff events. The *rc* is computed as the ratio of total runoff to event rainfall and illustrates the quantities of rainfall released from the catchment and stored by vegetation, soil moisture, and percolation into the subsurface. The *ts* is the total duration of one rainfall-runoff event.

The event time scale determines the hydrograph and duration of an event. Longer and shorter *ts* values denote slow com-

ponent runoff (e.g., subsurface or through flow) and fast runoff generation (e.g., surface overflow), respectively. Most rainfall-runoff events in the semi-arid and humid climates of the Loess Plateau are a mixture of fast and slow runoff generation patterns. The *rt* is another important rainfall-runoff event variable and represents the total runoff duration until peak discharge (how fast the peak appears). In a complicated runoff generation process, earlier peaks (i.e., short event rise time) are the main contributors to fast processes and longer recession times. The *peak*, the biggest discharge of one identified runoff event, provides information





**Table 3.** Overview of rainfall-runoff event characteristics and indicators of pre-event catchment state used in this study

|  | Abbreviation | Unit | Data Source |
|---|---|---|---|
| **Event characteristics** | | | |
| Event runoff coefficient | *rc* | dimensionless | Event separation |
| Event times scale | *ts* | hours | Event separation |
| Event rise time | *rt* | *h* | Event separation |
| Peak discharge | *peak* | m³/s | Event separation |
| **Rainfall characteristics** | | | |
| Rainfall event volume | *Pvol* | mm | Calculated by rainfall data |
| Maximum intensity of rainfall event | *Pint* | mm/h | Calculated by rainfall data |
| **Indicators of pre-event catchment state** | | | |
| Antecedent soil moisture | *Pa* | mm | Calculated by rainfall and evapotranspiration data |
| 10-day antecedent rainfall | *Pant10* | mm | Calculated by Rainfall data |
| Maximum catchment water storage | *Wm* | mm | Calculated by runoff data |
| Baseflow at the beginning of the event | *qbase* | m³/s | Event separation |

on the magnitude. Rainfall event characteristics (e.g., volume and intensity) and catchment wetness (e.g., antecedent rainfall, soil moisture, catchment water storage, and base flow) were studied to understand dominant long-term runoff events.

Table 3 shows the metadata of the indicators. The effect of basin wetness on runoff processes was studied with a similar procedure.

### 3.3.2 Monthly deviation of event characteristics

Rainfall is primarily concentrated in summer. Thus, runoff event periods are: June, July, August, and September. Monthly event characteristics were quantified by the value of deviation computed by the long-term monthly and annual mean values. Monthly rainfall and catchment wetness characteristics were analyzed to determine the relationship between event characteristics and drivers.

### 3.3.3 Long-term changing of event characteristics

Mann-Kendall non-parametric tested mean monthly values were used to estimate the possible long-term variability of event characteristics in typical Loess Plateau catchments. The Mann-Kendall test is usually used for hydrological change analysis (e.g., water-balance-based runoff coefficient, hydrological extremes, analysis of hydro-climatic, and water quality treads). Because of the rainfall-runoff characteristics, the non-parametric Sen's slope was chosen to estimate the magnitude of the trend. Land use and land cover in the five catchments were also analyzed to investigate the effect of local land conditions on

hydrological trends, and the total area of land use change in the catchments from 1992–2015 was studied.



## 4  Results

### 4.1  Runoff event separation

We tested various base separation methods to select a suitable methodology for further event identification. Parametric digital filter (e.g., Chapman, Chapman & Maxwell and Eckhardt), HYSEP (P&H- fixed window), and nonparametric simple smooth-
ing methods were compared. Because of the absence of 'true base flow' for comparison, the performance of various methods was evaluated by potential runoff events (Figure 5) and base flow index (*BFI*) calculated by the ratio of Q90 to Q50 (Figure 6).

Base flow values in the Loess Plateau are small compared with the peak flow. The peak discharge value is always 100~150 times larger than that of the base flow. The base flow estimations from the Lyne & Hollick, Chapman, Chapman, Maxwell, and Eckhardt digital filters (Figure 5) were close to the actual results. Compared with the recession process, the Lyne &
Hollick method produced values closer to those of the observations. The runoff process of the other methods (except IH) were complicated. Figure 6 shows the BFI value distribution. The mean values of the Lyne & Hollick, P&H -fixed window, Chapman & Maxwell, Chapman, Echhardt, and IH were 0.52, 0.52, 0.51, 0.50, 0.48, and 0.47, respectively. The calculated results fluctuated by 0.47~0.52 (with 9.6% of the highest values). The simple IH method performed the best in identifying the beginning of events in the three catchments. The BFI was similar to that of other methods, and thus, the IH method was
selected for base flow separation for the identification of a single rainfall-runoff event.

### 4.2  Rainfall-runoff characteristics

A total of 340 events were chosen from the runoff series in the five Loess Plateau sub-catchments from 1960–2014. The runoff events, in the different runoff periods, for each catchment varied from 15–125 (Figure 7a). Because of the runoff series limitations, only 15 rainfall-runoff events were selected for the Zulihe basin from 2006–2014. The Jingle and Gushanchuan
basins had 98 and 125 events, respectively. In each catchment, 2~3 runoff events were selected every year. The runoff coefficient ($rc$) represents the amount of rainfall that becomes runoff. The average $rc$ values in the five catchments were close to 0.2 (Figure 7b). The $rc$ values of Chabagou, Gushanchuan, and Jialuhe were larger than those of Jingle and Zulihe. However, the Jingle and Zulihe catchments featured 11 runoff events with $rc$ greater than 0.4. These results were consistent with precipitation intensity characteristics ($pint$) (Figure 7c). $pint$ values are larger in the Chabagou, Gushanchuan, and Jialuhe catchments than in Jingle
and Zulihe beyond 17 mm/h. However, time scales ($ts$) ( Figure 7d) of the Jingle, Zulihe, and Jialuhe catchments were larger than those of the other catchments (every event approached 100 h). Chabagou and Gushanchuan had greater rainfall intensity and shorter event time scales.

Figure 8 shows the relationship between the summation of antecedent impact precipitation (representing catchment wetness) and rainfall and runoff volume to identify rainfall-runoff characteristics in different regions of the Loess Plateau and reveals
a nonlinear relation between the three variables. In each catchment, an exponential function was used to fit the runoff volume for antecedent impact precipitation and event rainfall. The Zulihe and Jingle catchments had rainfall-runoff features similar to those of Chabagou, Gushanchuan, and Jialuhe. Under the same catchment wetness and precipitation, Chabagou, Gushanchuan,

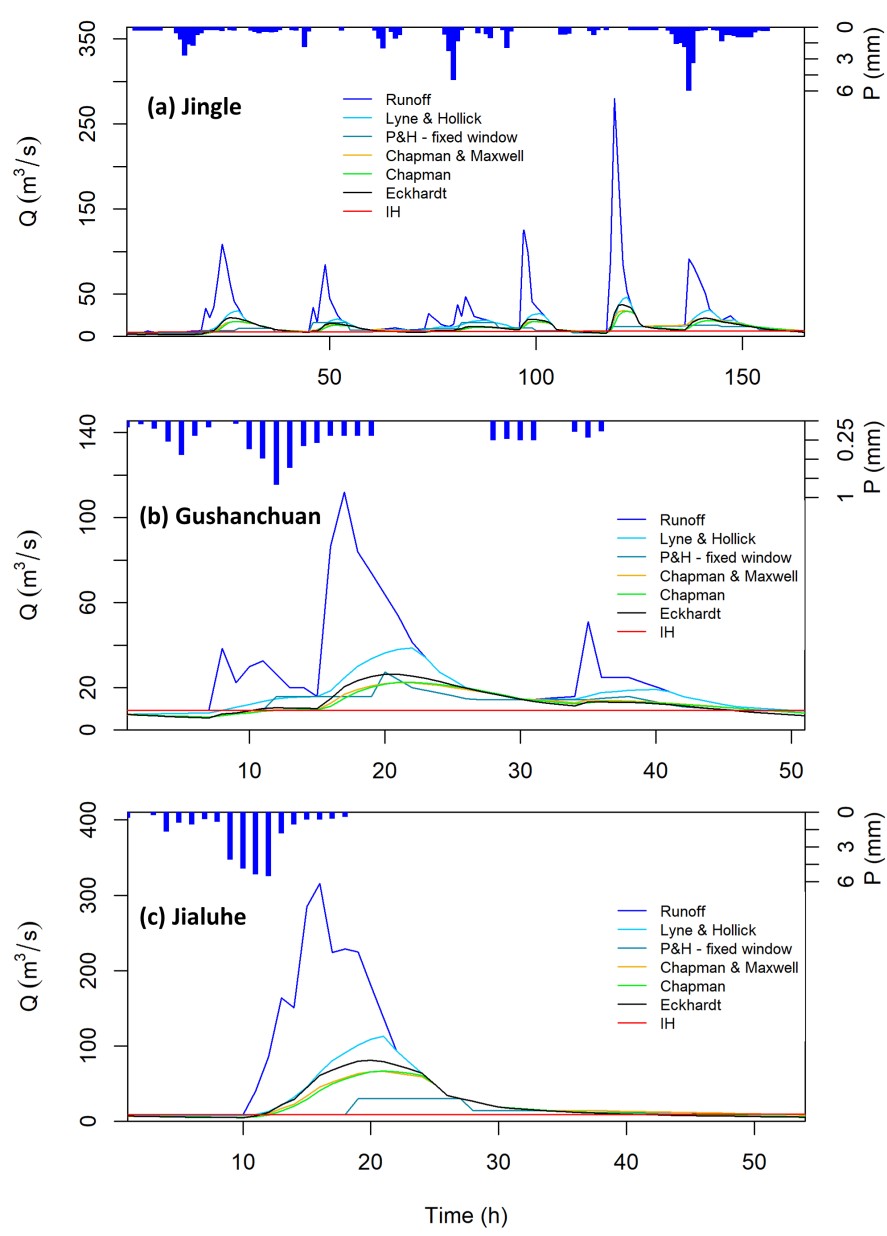

**Figure 5.** Partial results of six base flow separation methods tested in three catchments (Jingle, Gushanchuan, and Jialuhe).



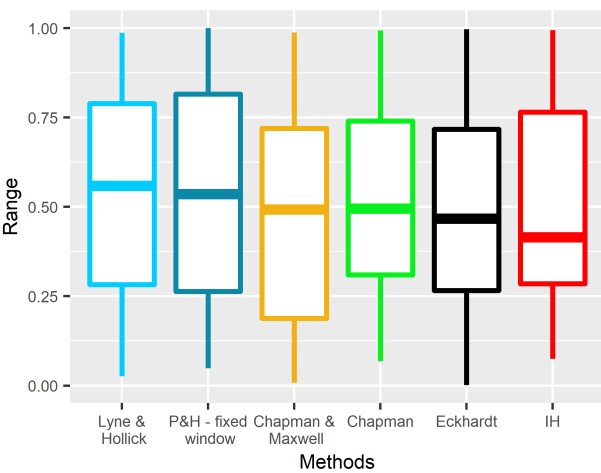

**Figure 6.** Base flow index (BFI) variation identified by different base flow separation methods for all catchments in the data set.

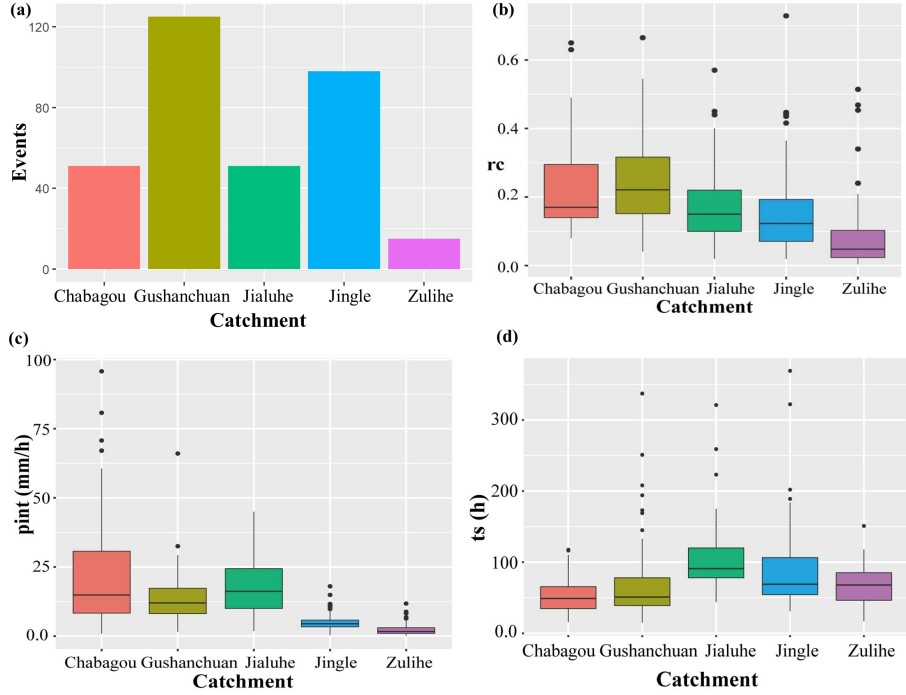

**Figure 7.** Rainfall-runoff event features with number of event (a), runoff coefficient (rc), precipitation intensity (pint), and runoff time scale (ts) in the five catchments.





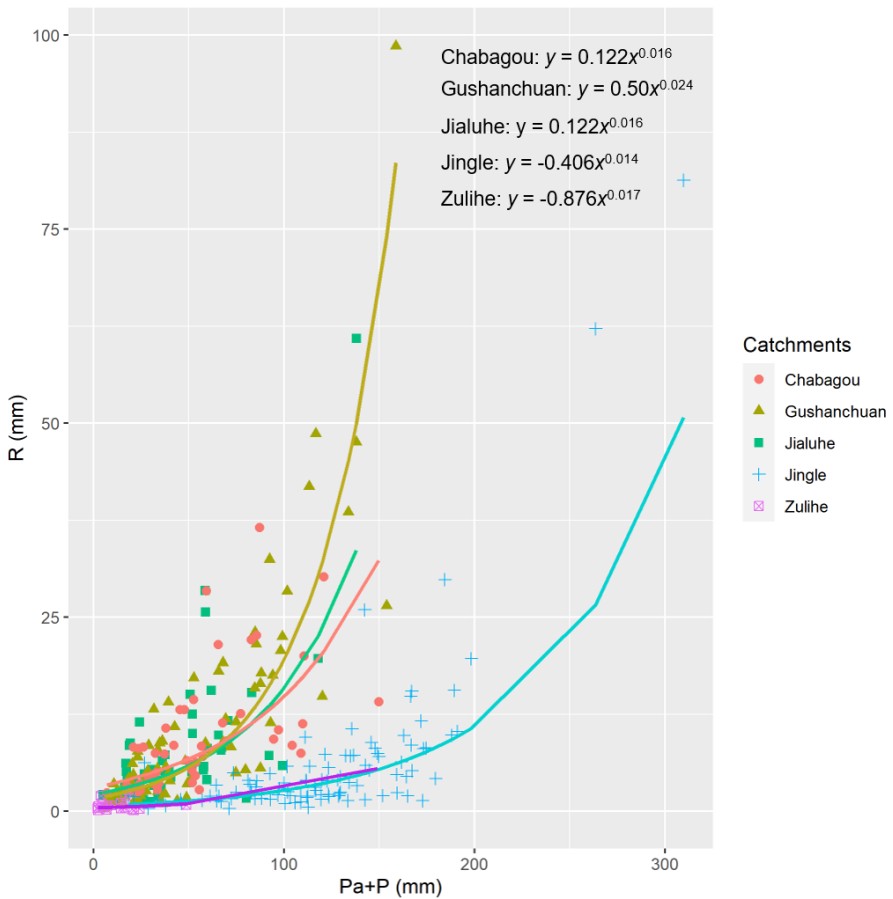

**Figure 8.** Nonlinear relationship of antecedent impact precipitation (pa), precipitation (P), and runoff volume (R) in the five catchments

and Jialuhe produced more runoff, and Gushanchuan was the highest catchment. These results agree with the above results (Figure 7).

## 4.3 Dominant runoff generation pattern

The hydrograph response, time source, and flow path of each event were analyzed for dominant runoff generation patterns with a diagnostic framework. Table 4 summarizes the runoff generation in the five catchments at different stages. The three primary periods of ecological measure implementation in the Loess Plateau were the 1970s, 1980s, and after the 1990s. The runoff series summarizes the dominant runoff generation pattern in each catchment by period. DOF runoff events appeared

after the 1980s in all the catchments, except Jialuhe. HOF and DOF combined runoff appeared at all time stages. These results illustrate that DOF runoff events could be formed in some rainfall-runoff events but were not the dominant catchment runoff process. The HOF and DOF combined events had the second highest occurrence. DOF runoff represented 15 of the 340 events



**Table 4.** Summary of runoff generation in five catchments over different stages (HOF: Hortonial overland flow; DOF: Dunne overland flow; Combinations: HOF and DOF).

| Periods | HOF | DOF | Combinations | Total |
|---|---|---|---|---|
| | | Jingle | | |
| 1971–1987 | 41 (80.39%) | 0 (0.00%) | 10 (19.61%) | 51 (100%) |
| 1988–1997 | 20 (66.67%) | 2 (6.67%) | 8 (26.67%) | 30 (100%) |
| 1998–2014 | 9 (52.94%) | 1 (5.88%) | 7 (41.18%) | 17 (100%) |
| 1971–2014 | 70 (71.43%) | 3 (3.06%) | 25 (25.51%) | 98 (100%) |
| | | Chabagou | | |
| 1960–1971 | 22 (79%) | 2 (7%) | 4 (14%) | 28 (100%) |
| 1972–1997 | 11 (73%) | 1 (7%) | 3 (20%) | 15 (100%) |
| 1998–2006 | 6 (75%) | 1 (12.5%) | 1 (12.5%) | 8 (100%) |
| 1960–2006 | 39 (76%) | 4 (10%) | 8 (14%) | 51 (100%) |
| | | Gushanchuan | | |
| 1965–1979 | 43 (71.67%) | 0 (0.00%) | 17 (28.33%) | 60 (100%) |
| 1980–1997 | 29 (58.00%) | 2 (4.00%) | 19 (38.00%) | 50 (100%) |
| 1998–2006 | 8 (53.33%) | 1 (6.67%) | 6 (40.00%) | 15 (100%) |
| 1965–2006 | 80 (64.00%) | 3 (2.40%) | 42 (33.60%) | 125 (100%) |
| | | Jialuhe | | |
| 1960–1971 | 22 (79%) | 2 (7%) | 4 (14%) | 28 (100%) |
| 1972–1997 | 11 (73%) | 1 (7%) | 3 (20%) | 15 (100%) |
| 1998–2006 | 6 (75%) | 1 (12.5%) | 1 (12.5%) | 8 (100%) |
| 1960–2006 | 39 (76%) | 4 (10%) | 8 (14%) | 51 (100%) |
| | | Zulihe | | |
| 2006–2014 | 10 (66.67%) | 1 (6.67%) | 4 (26.67%) | 15 (100%) |
| | | Total | | |
| 1960–2014 | 238 (70.00%) | 15 (4.41%) | 87 (25.59%) | 340 (100%) |

(4.41%) in the five catchments. HOF was the dominant runoff process in the five catchments. In the Jingle and Gushanchuan catchments, its proportion decreased from the 1970s to the 2000s.

Table 5 shows the 15 DOF-dominant runoff events selected from 1960–2014 with weights of rainfall-runoff characteristics. Only 3~4 runoff events were identified with the DOF-dominant runoff process in each catchment, as excess-saturation runoff is rare in the Loess Plateau. However, the hydrograph response (hydrograph symmetry, rainfall duration, and discharge rise time), time source (catchment water shortage and runoff coefficient), and flow path (surface soil structure, land use type, and relevance with intensity or volume) analyses (4 reveal 15 possible DOF-dominant runoff events. The final evidence weight values ($W_{MC}$) were below 0.5 (Table 5). The hydrograph response weights ($W_{HR}$) were above 0.5; the weight of flow path





**Table 5.** Overview of evidence weights for Dunne overland flow (DOF) in the five catchments. The three stormflow characteristics (hydrograph response, flow path, and time source) are shown.

| - | Event weight | hydrograph response $(W_{HR})$ | Flow Path $(W_{FP})$ | Time Source $(W_{TS})$ | Final Evidence Weight $(W_{MC})$ | Dominant Runoff Processes |
|---|---|---|---|---|---|---|
| Event No. | Catchment | - | - | - | - | - |
| 19950831 | Jingle | 0.5 | 0.5 | 1 | 0.25 | DOF |
| 19960801 | Jingle | 0.5 | 1 | 0.5 | 0.25 | DOF |
| 20100810 | Jingle | 1 | 1 | 0.5 | 0.25 | DOF |
| 19630826 | Chabagou | 1 | 0.5 | 1 | 0.5 | DOF |
| 19780807 | Chabagou | 1 | 0.5 | 0.5 | 0.25 | DOF |
| 19910607 | Chabagou | 1 | 0.5 | 0.5 | 0.25 | DOF |
| 20060812 | Chabagou | 1 | 0.5 | 0.5 | 0.25 | DOF |
| 19950728 | Gushanchuan | 0.5 | 1 | 0.5 | 0.25 | DOF |
| 20010816 | Gushanchuan | 0.5 | 0.5 | 1 | 0.25 | DOF |
| 20050812 | Gushanchuan | 0.5 | 1 | 0.5 | 0.25 | DOF |
| 19670731 | Jialuhe | 0.5 | 1 | 0.5 | 0.25 | DOF |
| 19710704 | Jialuhe | 1 | 0.5 | 1 | 0.5 | DOF |
| 19940822 | Jialuhe | 0.5 | 1 | 0.5 | 0.25 | DOF |
| 20010816 | Jialuhe | 1 | 0.5 | 0.5 | 0.25 | DOF |
| 20140816 | Zulihe | 1 | 0.5 | 0.5 | 0.25 | DOF |

$(W_{Fp})$ and time source $(W_{TS})$ were lower than that of $W_{MC}$. Thus, due to the sporadic occurrence of DOF, the HOF runoff process was dominant.

## 4.4 Temporal dynamics of event runoff response

A quantitative analysis of the event runoff response and subjective analysis of the dominant runoff generation process in each
catchment were used to test the above results. Land use variability, event-to-event runoff characteristics, and monthly and long-term event characteristics are studied in this section.

### 4.4.1 Land use variability of catchments

Figure 9 shows the distribution of land use in 1992. The Zulihe and Jingle catchments had similar land use types with large areas of grass and herbaceous cover. Chabagou and Jialuhe had vast areas of cropland. Gushanchuan was primarily composed
of natural vegetation. Figure 9 shows relative land use change from 1992–2015 in cropland, herbaceous land, and grassland in





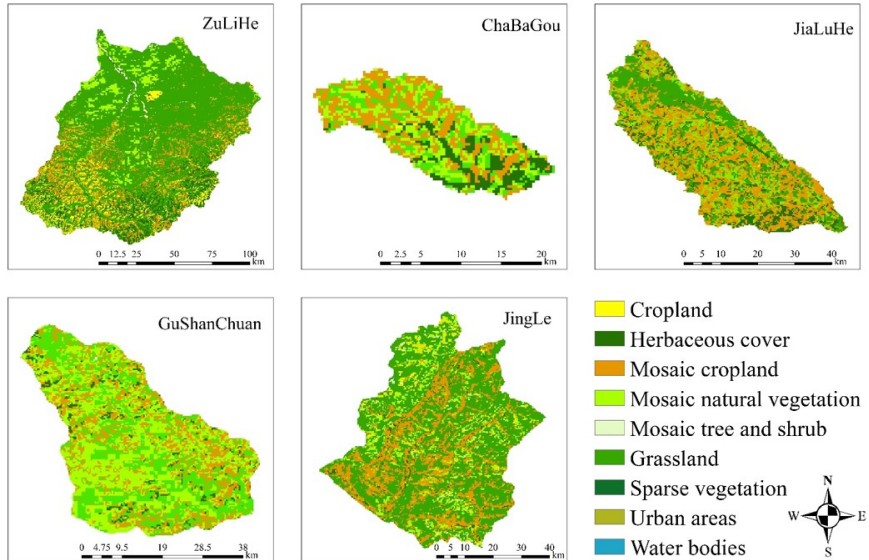

**Figure 9.** Land use distribution in the five catchments in 1992. (Land use data obtained from cci)

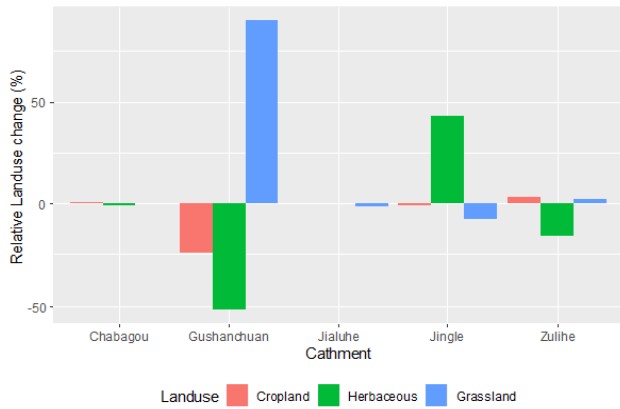

**Figure 10.** Relative land use change between from 1992—2015 in for cropland, herbaceous land, and grassland of in the five catchments

the five catchments. Grassland significantly increased in Gushanchuan as cropland and herbaceous land decreased. Herbaceous land increased in Jingle, and land use changes in Gushanchuan, Zulihe, and Chabagou were not significant.

### 4.4.2 Event-to-event runoff characteristics

The relationships of runoff event characteristics (event runoff coefficient ($rc$), time scale ($ts$), rise time ($rt$), peak discharge ($peak$)), and rainfall volume ($Pvol$) with the pre-event catchment state indicators (antecedent impact precipitation ($Pa$), pre-






event base flow ($base$), and catchment water storage ($Wm$) were studied (Figure 11) to gain insight into rainfall-runoff characteristics. We studied how the rainfall properties and pre-catchment conditions alter the event runoff coefficient, time scale, rise time, and peak discharge at the catchment scale. The relationship between rainfall and runoff is generally nonlinear and complicated.

Runoff coefficient values ($rc$) increased higher than 0.2 with precipitation volume ($Pvol$) for $Pvol$ values less than 100 mm. However, these values remained close to 0.2 for $Pvol$ larger than 100 mm. The majority of $rc$ values in the DOF and combination runoff generation processes were larger than those of the HOF. An antecedent impact precipitation $Pa$ value of 62.5 mm separated the $rc$ distribution laws. The relationship between base flow ($qbase$) and $rc$ was significant, but $rc$ and $Wm$ were unrelated. Larger $Wm$ mainly corresponds to DOF-dominant events.

The event time scale ($ts$) increased with $Pvol$ but was unrelated to $Pa$, $qbase$, and $Wm$; $ts$ values were determined based on rainfall characteristics. However, event rise time ($rt$) values increased as $Pa$, $qbase$, and $Wm$ increased. The special characteristic of the Loess Plateau caused peak discharge ($peak$) and $Pvol$ to be significantly related. Thus, rainfall characteristics and catchment wetness primarily control the $rc$ and $ts$ characteristics of runoff.

### 4.4.3 Monthly changing of event characteristics

The majority of runoff events occurred from June–September. The monthly deviations (%) of rainfall-runoff characteristics and pre-event wetness in each catchment were studied from June–September (Figure 12). The values of these characteristics in July and August were generally larger than those in June and September, illustrating significant monthly variation. In the five catchments, the values of $rc$ were larger in August and July. In almost all catchments, $ts$ was greater in August, whereas $rt$ was greater in June. The values of $peak$ did not have obvious deviations (compared to $rc$, $ts$, and $rt$) from June to September

but were lower in September. The values of $Pvol$ and $pint$ were larger from June–September. Except in the Zulihe catchment, $Pa$ values increased from June–September. There were no significant monthly differences in $Wm$. Runoff ($rc$, $ts$), and rainfall ($Pvol$ and $pint$) characteristics showed monthly variations. Catchment wetness increased from June–September.

### 4.4.4 Long-term trends of event characteristics

Figures 13 and 14 show a changing runoff event response and the relationship between climate and conditions in the catch-
ments. The value of Sen's slope was used to test runoff response from 1960–2014 and was computed in June, July, August, and September. The $ts$ was first studied to determine long-term change in event duration (Figure 13) and decreased in the Jingle catchment. Land use in the Jingle catchment changed little, but $Pvol$ and $pint$ increased in July and August; $ts$ in Zulihe increased in July, August, and September and remained steady in June. Land use in Zulihe changed by 2~4%. $ts$ changes in the Chabagou, Gushanchuan, and Jialuhe catchments near the Loess Plateau were significantly different, $ts$ increasing in
July, increasing in August, and decreasing from June–August in Jialuhe, Gushanchuan, and Chabagou, respectively. Rainfall characteristics ($Pvol$ and $pint$) increased in July in the five catchments. The wetness indicator $Pa$ also increased every month in all the catchments.





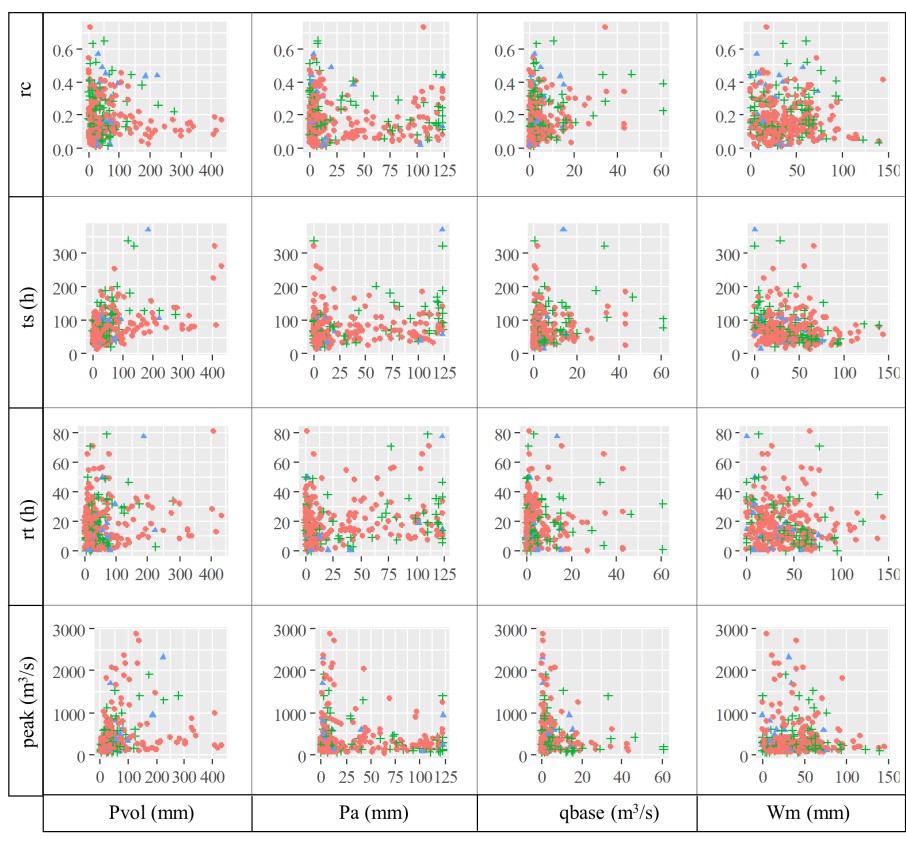

**Runoff generation pattern** • HOF ▲ DOF + Combinations

**Figure 11.** Relationship between runoff event characteristics (event runoff coefficient (rc), time scale (ts), rise time (rt), peak discharge (peak)), rainfall volume (Pvol), and pre-event catchment state indicators (antecedent impact precipitation (Pa), pre-event base flow (base), and catchment water storage (Wm)).

Figure 13 shows long-term event runoff coefficient ($rc$) changes; $rc$ is a comprehensive value for representing runoff response and decreased in July in all catchments. The value of Sen's slope was the highest Jialuhe. $rc$ increased in Chabagou in June and September and decreased continuously in Gushanchuan, Jialuhe, Zulihe, and Jingle. $Pvol$ and $pint$ decreased in July and August, which had higher rainfall-runoff deviation (Figure 12), but the decreasing trend became less pronounced. Thus the runoff trend in the catchments was slowing, and catchment wetness ($Pa$ and $Wm$) was increasing (Figure 13). Decreasing rainfall and increasing wetness reduced the amount of runoff.

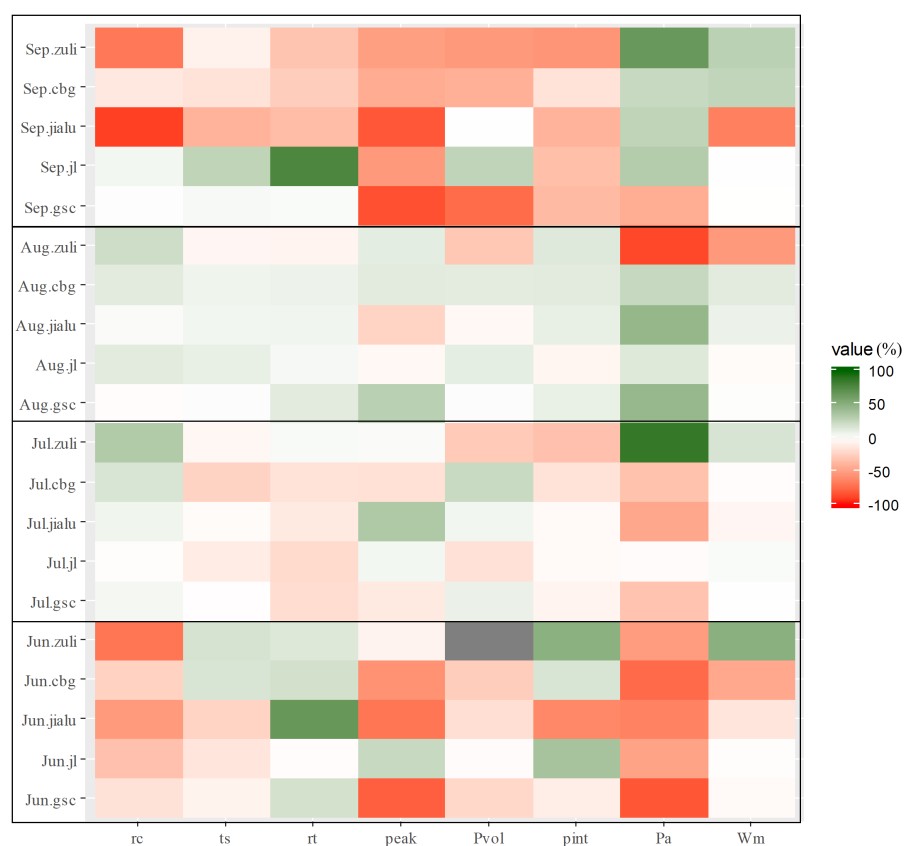

**Figure 12.** Monthly variation of event characteristics, rainfall properties, and pre-event wetness. These values were computed as monthly deviations from the mean annual variables, and zuli, cbg, jialu, jl, and gsc are abbreviations for the five catchments. Monthly deviation is shown from June–September.



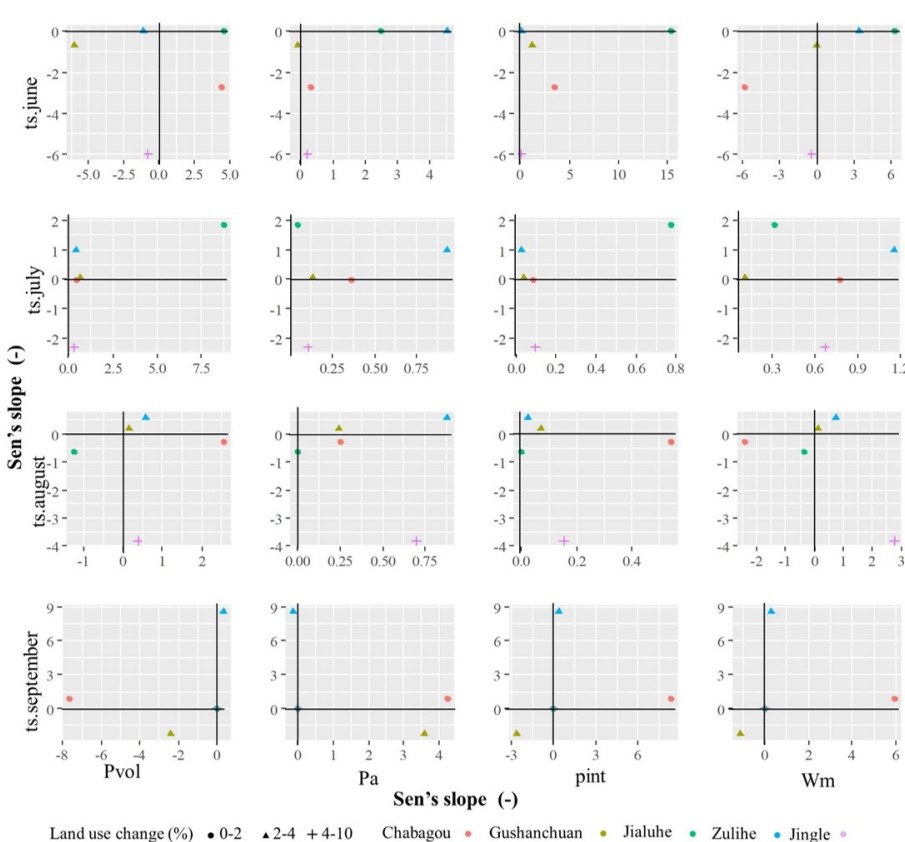

**Figure 13.** Long-term event time scale (ts) changes. Monthly (ts) trends were plotted against monthly rainfall properties and pre-event catchment wetness trends, which were quantified with Sen's slope, which can be interpreted as the mean annual (percent) change in the long-term mean value of the variable. Each marker represents a catchment belonging to a specific natural region differentiated by color. The shape of the marker indicates the relative land use change in the catchment from 1992–2015.



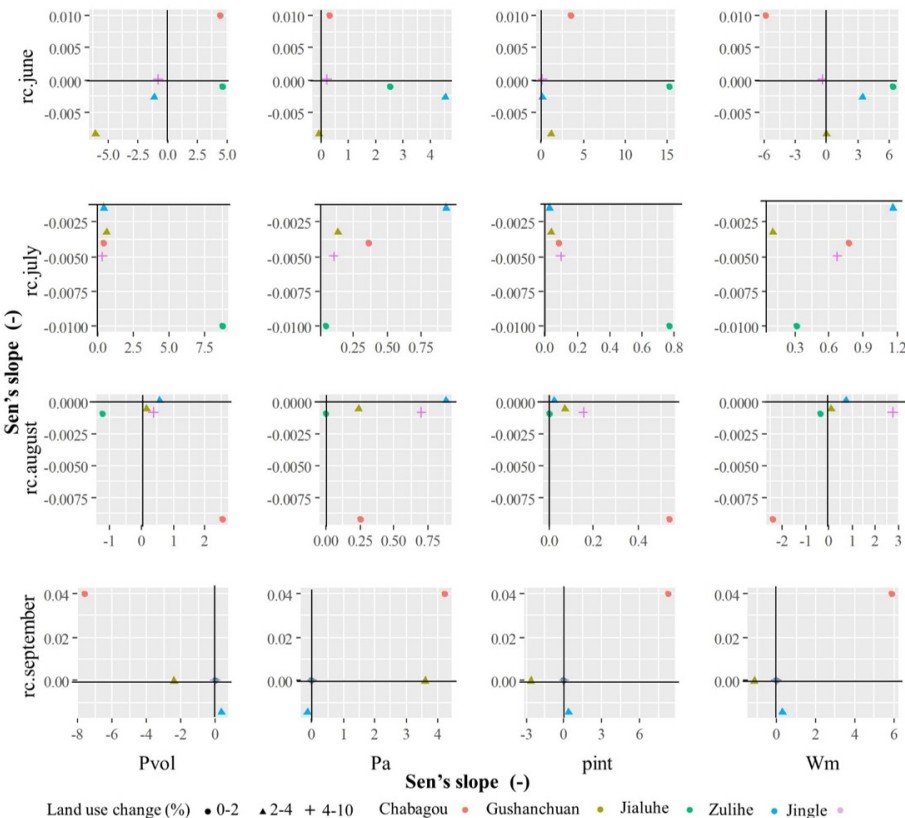

**Figure 14.** Long-term event runoff coefficient (rc) changes. Monthly rc trends were plotted against monthly rainfall properties and pre-event catchment wetness trends. Trends were quantified by referring to the relative Sen's slope, which is the mean annual percent change of the long-term mean value. Each marker represents a catchment belonging to a specific natural region differentiated by color. The shape of the marker indicates the relative land use change from 1992–2015.

## 5 Discussion

The event identification method was used to determine the mechanisms of catchment runoff generation. This method considered base flow separation and rainfall attribution. With the identification of event runoff, the temporal dynamics and characteristic response of large-scale runoff generation could be studied. This research was applied to gain insight into the catchment scale runoff process used by many hydrologists (Dunne, 1978; Seibert et al., 2016; Tarasova et al., 2018b, a). In this study, six base flow separation methods were tested. Finally, the smoothing approach (IH, 1980) was chosen for base flow separation

because it better identified the starting points of runoff events and was simple for the five catchments. The semi-arid regions of the Loess Plateau provide a small base flow with low rainfall duration, producing quick flow into rivers after rainstorms. The simple smoothing approach is more appropriate in Loess Plateau catchments than more complicated base flow separation methods (e.g., parameter digital filtering, recursive digital filtering, and HYSEP methods).





Rainfall in the Loess Plateau usually lasts a few hours or less in a day (Luo et al., 2013). Because of the arid environment,
scarce rainfall would not cause a sharp increase in discharge. Thus, medium and large rainfall events were selected, and 340
runoff events were identified in the five catchments, which did not have large-scale water conservancy project constructions
but had implemented ecological protection measures. The representative catchments were Zulihe in the upper reaches of the
Yellow River; Chabagou, Gushanchuan, and Jialuhe in the primary sand-producing area of the Yellow River; and Jingle in the
upper reaches of the Fen River Basin. The characteristics of the underlying surface of the basin of the catchments represented
the primary geographical characteristics of the Loess Plateau. The study periods were not consistent in all the catchments
because of the absence of runoff or rainfall data.

Hourly hydroclimatic series were used to identify rainfall-runoff events. Hourly resolution ensured the identification of
runoff events shorter than 1 day but with large rainfall volume. The five catchments are mesoscale basins larger than 100 km$^2$.
The data was used to study the event hydrograph response, time source, and flow path to analyze the event runoff generation
process. Additionally, the data series for Chabagou, Gushanchuan, and Jialuhe were from the 1960s to the 2000s. These long-
term series are sufficient to study monthly and long-term event dynamics and are suitable for our catchment location and
rainfall-runoff data. Moreover, the $rc$, $ts$, $rt$, $peak$, $Pvol$, and $pint$ rainfall-runoff characteristics can be computed with the
hourly data and used widely by hydrologists for researching hydroclimatic catchment features.

Obtaining the real values of each runoff event (Hewlett and Hibbewitrt, 1967) is very difficult, and real $peak$, $ts$, and
runoff generation patterns are vague in rainfall-runoff processes. However, event identification and diagnostic runoff generation
procedures were applied to the 340 runoff events and provided insight into the dynamics of the hydrology processes of the Loess
Plateau allowing the study of long-term runoff response in a changing environment.

The dominant runoff generation processes were analyzed with a runoff generation classification framework. Surface storm
runoff can be produced by HOF and DOF flow patterns. When the rainfall intensity exceeds soil infiltration, HOF (flow path
1, Figure 15A) appears (Horton, 1933). Deep ground water is not recharged by infinite rainfall (flow path 2, Figure 15A),
which does not contribute to surface runoff but may form a shallow subsurface flow (flow path 3, Figure 15A) (Dunne, 1978).
Whipkey (1965) found such that shallow subsurface flow traveled in soil layers through river channels before reaching the
groundwater zone. Throughflow near-storm channels (flow path 4, Figure 15) and saturated overland flow (flow path 5, Figure
15) occur on saturated surfaces.

The hydrograph is determined by the HOF and DOF runoff patterns (Dunne, 1978), and the characteristics of the hydrograph
reflect the rate of soil infiltration, runoff time source, and flow path in a runoff event. High peak discharge and quick recession in
a short time (Figure 15B) denote dominant HOF runoff with low soil infiltration. Lower peak discharge and a slower recession
limb denote DOF dominant runoff with high soil infiltration (Figure 15B). Throughflow lasts longer than infiltration-excess
overland flow. Thus, the hydrograph response and other factors indicated that runoff source time and flow path should be
considered for runoff generation classification (Figure 4). The results showed that HoF is dominant in the majority of runoff
events in the Loess Plateau. The DOF and combination-dominant runoff events occurred after the 1980s.

Our results are consistent with those of other studies on Yellow River discharge and sediment change (Yao et al., 2009, 2013,
2015; Yao and Jiao, 2016) that used long-term climatic characteristic, runoff, and land surface data series to find that climate





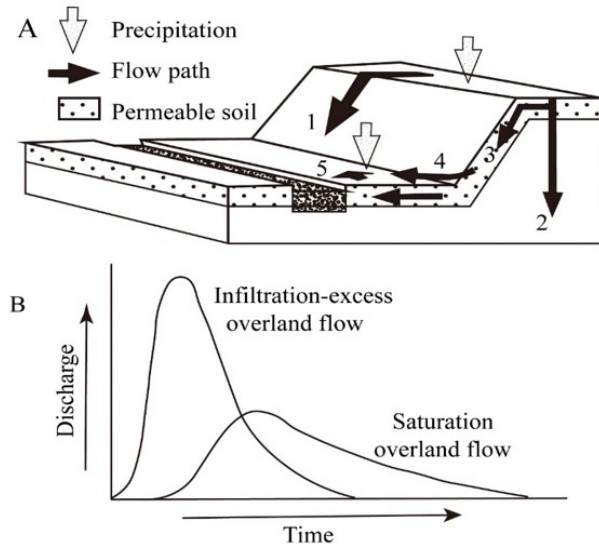

**Figure 15.** Runoff path and characteristics of hydrographs in a rainfall-runoff event.

and human activities were the primary factors affecting Yellow River runoff changes. The source of runoff formation may

transform to humid regional flow patterns. Numerous rainfall-runoff experiments have been conducted on the Loess Plateau (Jing, 2008; Sanda et al., 2014; Suarez et al., 2015), but the experimental scale limited insight into the dynamics of long-term spatiotemporal runoff changes. In this study, runoff generation at the catchment scale was studied to supplement previous studies on water and sediment changes in the Yellow River.

Runoff event response was analyzed to quantitatively study rainfall-runoff process dynamics in the Loess Plateau. Runoff

events were controlled by rainfall properties (i.e., intensity and volume). However, the relationship between runoff and rainfall characteristics is complicated, and different features had different runoff generation patterns. DOF-dominant runoff events had a strong relation with precipitation amount (Figure 11). However, HOF-dominant runoff events were weakly related to many factors. Thus, runoff generation mechanisms in the Loess Plateau are complex and changeable. Event runoff coefficients in the five catchments were higher in July and August, indicating maximum runoff in those months. The catchment wetness $Pa$

indicator changed monthly. Pre-event saturation also greatly affected runoff generation in the Loess Plateau. Therefore, August and July are the most important months for runoff formation.

The characteristics of monthly events control month-specific long-term changes (Figure 13 and 14). Precipitation volume and intensity mostly increased in the five catchments. According to Fu et al. (2017), this long-term change might have been caused by climate change in the Loess Plateau. Changes in precipitation amount and intensity do not explain the conversion

of runoff generation patterns. However, changes in land use and the transformation of precipitation might accelerate runoff generation changes. In this study, DOF runoff events (15 events) occurred in all five catchments, but are supposed to occur only in humid conditions. Catchment storage capacity increased after the increasing the forest land and grassland and constructing





terraces and check dams, which increased soil infiltration. Additionally, macropore flow caused by vegetation growth increased soil infiltration capacity (Alaoui et al., 2011; Gerke et al., 2015; Dusek and Vogel, 2016). Saturation excess runoff occurs for

catchments with low storage capacity during long rainfalls, an unusual situation in the study area. The five catchments are characterized by runoff generation from excess infiltration. Land use changes (more than 2% in each catchment) and climatic characteristics might explain hydrological inconsistencies in the Loess Plateau, and the runoff response can be affected only if the 15% of the catchment area has changed (Martin et al., 2012). We only studied land use from 1992–2015 because of data limitations; however, the land surface changed sharply in the 1980s, and the event time scale, rise time, and runoff coefficients

were mostly affected by land use changes. Runoff coefficient values decreased in the five catchments, but the manner in which land use modified the hydrological response is still poorly understand (Sawicz et al., 2014). We speculated that land use change (large-scale vegetation restoration) affects long-term trends in runoff characteristics by considering runoff generation and response. Thus, the establishment of hydrological models, water resource management, and ecological measures in the Yellow River Basin should carefully consider the conversion of runoff generation mechanisms. Nonetheless, this study only

covered five catchments in the middle reaches of the Yellow River, which may not be representative of the vast Loess Plateau. We anticipate further research on the runoff generation mechanisms in the Yellow River Basin.

## 6 Conclusions

In this study, we studied temporal dynamics of runoff generation patterns and runoff characteristics collected from hydrological time series to to explain reduction of runoff by catchment processes of runoff generation. By comparing six base flow separation

methods, the nonparametric simple smoothing method was suitable for Loess Plateau. With proposing a framework procedure to classify dominant runoff pattern allows for comparison of dominant runoff generation patterns on temporal scale. The dominant runoff generation could be revealed by temporal dynamics of rainfall-runoff characteristics. Mostly, because of changes of rainfall properties, catchments runoff of dominanted by infiltration or saturation excess are more easily to change in Loess Plateau. we analyzed 340 identification events in the five catchments. The runoff coefficient in the Zulihe catchment was

lower than that in the other catchments and decreased in July and August, whereas the event time scale increased. The other catchments were similar with Zulihe catchment with declining trend of runoff coefficient and increasing trend of time scale. In these catchments rainfall is stored more by flow paths changing because of increasing vegetation and lead to changeable water and sediment to rivers. An analysis of runoff generation patterns indicated that infiltration-excess runoff remains predominant but decreased in the last 10 years, because of increased cropland and forest that increased the infiltration rate. Therefore,

land use changes reduced the runoff coefficient for water discharge and decreased sediment content. Understanding temporal dynamics of runoff characteristics and dominant runoff generation patterns is crucial for the management of water resources, ecological restoration, control of river sediment and prevention flood hazard.



*Acknowledgements.* The authors would like to acknowledge the financial support received from Projects of National Natural Science Foundation of China (51979250), National key research priorities program of China (2016YFC040240203), Key projects of National Natural Science Foundation of China (51739009) and Key Research and Promotion Projects (technological development) in Henan Province (202102310587).




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
