# Peer review of "Rainfall-runoff processes in the Loess Plateau, China: Temporal dynamics of event rainfall–runoff characteristics and diagnostic analysis of runoff generation patterns"

_Hydrology and Earth System Sciences, 2020_

## Referee Comment (RC1) · Anonymous Referee #1 · 1 Dec 2020

This research examined the runoff generation mechanism at the catchment scale to understand thechange in runoff generation based on long-term series monitoring data, which could make fundamental progress in the prevention of flooding hazard and the management of water resources in the Yellow River. The paper presents the research in a logical format and the methods are clear. However, the paper presented in its current form has a number of issues that do not make it publishable at this stage. Therefore, I would recommend make major revision based on the following suggestions: 1. The novelty of this paper, in particular the method used is limited in the current version

and should be emphasized. 2.Why chose the 5 catchments for analysis and the representative of the 5 selected catchments should be clarified. 3.The analysis of effect of land use and climate change on the runoff pattern was too simplified, more detailed quantitative analysis is needed to enhance our understanding. 4. How to consider the hetergeneous of underlying surface and rainfall in the 5 catchments, and detailed hydrological parameters such as infiltration rate is lacked. 5. Line 315 "The effect of antecedent soil moisture and rainfall on the runoff was not considered "Thus, medium and large rainfall events were selected, and 340 runoff events were identified in the five catchments, which did not have large-scale water conservancy project constructions but had implemented ecological protection measures". This was not exactly true as many check dams have been constructed in the Chabagou, how the runoff was influenced by these anthropogenic engineering measures lacked correspond analysis. 6. Ecological recovery has double-side effect on the runoff, is this the reason for runoff reduction in Yellow River or the shift from HOF to DOF need more detailed explaination and evidence.

Other minor comments Grammar and spelling mistakes such as P7 Line 143 "developmen".

---

## Referee Comment (RC2) · Anonymous Referee #2 · 5 Jan 2021

The manuscript "Rainfall-runoff processes in the Loess Plateau, China: Temporal dynamics of event rainfall–runoff characteristics and diagnostic analysis of runoff generation patterns" reported a study concerning the mechanism changing of overland flow generation in Loess Plateau catchments. The method in which the authors identified dominant runoff generation process in the Loess Plateau catchments is confused and not reliable. There is a lack of quantitative approach of identifying the mechanism changing of runoff generation and this made the reliability of the conclusion unsatisfactory.

[Figure]

On the other hand, the characteristics of rainfall–runoff processes are highly related to overland flow movement and concentration in catchments, particularly at the small and medium spatial scales. How did the authors deal with this? How to separate the effect of runoff-generation mechanism changing and the effect of overland flow movement condition changing led by LUCC?

Specific comments: 1. Line 5-10: are the mechanisms of runoff generation in these Loess Plateau catchments really changed? Any field evidence or reference provided in the main text? 2. Check the citation format of references in introduction. For example in line 22, the format of '(Jiongxin, 2005)' is wrong. 3. Line 98: why did the authors choose these spatial scales (100~10000 km2) and why these catchments? Please give the reason. 4. Line 110: why did the authors employ this approach to estimate antecedent soil moisture for rainfall-runoff events? Is there any reference? 5. Line 116: why there three steps? Please give references to make it clear. 6. Section 3.2: it seemed that this section introduced a method of qualitatively identifying saturation-excess overland flow. However, there is no reference concerning this method. Is there any verification of this method? Did the authors have any field study to check the result of this method? 7. Section 3.2: using the method above, how did the authors identify the combination of infiltration-excess overland flow and saturation-excess overland flow in catchments? It looks like the authors used the 'probability' of saturation-excess overland flow introduced in the method to represent the combination of the two types of overland flow. Is this verified and reliable? 8. Line 207: is there any research or measured data to support this point? 9. Line 217-220: why you choose these rainfall-runoff events. 10. Check dams have been widely and massively constructed in the Loess Plateau, modifying rainfall-runoff processes in catchments (e.g., the Chabagou catchment in this study). How did the authors isolate their impacts on rainfall-runoff processes from LUCC impacts?

---

## Author Comment (AC1) · 1 Feb 2021

General comments:

This research examined the runoff generation mechanism at the catchment scale to understand the change in runoff generation based on long-term series monitoring data, which could make fundamental progress in the prevention of flooding hazard and the management of water resources in the Yellow River. The paper presents the research in a logical format and the methods are clear. However, the paper presented in its

current form has a number of issues that do not make it publishable at this stage. Therefore, I would recommend make major revision based on the following suggestions.

Answer: Thank you for your encouragement. We have taken the time to think through all of your comments and carefully revised the manuscript as you suggested. Thank you for your valuable suggestion to improve the quality of the manuscript.

Specific points:

(1) The novelty of this paper, in particular the method used is limited in the current version and should be emphasized.

Answer: Yes, the method was not clearly described and it would be particularly emphasized in next version.

(2) Why chose the 5 catchments for analysis and the representative of the 5 selected catchments should be clarified.

Answer: Yes, the representative analysis of the five selected catchments would be added in next version. These five basins have been used in many studies and they are special for the Yellow River Basin to understand rainfall-runoff changing.

(3) The analysis of effect of land use and climate change on the runoff pattern was too simplified, more detailed quantitative analysis is needed to enhance our understanding.

Answer: Yes, more quantitative analysis of land use and climate change are needed for understanding runoff generation changing.

(4) How to consider the heterogeneous of underlying surface and rainfall in the 5 catchments, and detailed hydrological parameters such as infiltration rate is lacked.

Answer: Yes, the five catchments have significantly heterogeneous of underlying surface and rainfall. In this study, characteristics of rainfall-runoff process and catchments were considered by spatial and temporal variability in each catchment. The lacked

hydrological parameters would be supplied in next version.

(5) Line 315 "The effect of antecedent soil moisture and rainfall on the runoff was not considered "Thus, medium and large rainfall events were selected, and 340 runoff events were identified in the five catchments, which did not have large-scale water conservancy project constructions but had implemented ecological protection measures". This was not exactly true as many check dams have been constructed in the Chabagou, how the runoff was influenced by these anthropogenic engineering measures lacked correspond analysis.

Answer: Yes, this place was not well described. Originally, the effects of check dams and terraced fields were not analyzed. Thank you for this suggestion. These anthropogenic engineering measures and influences on runoff generation would be detailed analyzed in next version.

(6) Ecological recovery has double-side effect on the runoff, is this the reason for runoff reduction in Yellow River or the shift from HOF to DOF need more detailed explanation and evidence.

Answer: Yes, it is a good suggestion. Ecological recovery has double-side effect on runoff generation. There is complicate relationship between ecological recovery and transformation of runoff partitioning. But I have collected some field experimental data in similar regions to explain this effect.

(7) Other minor comments Grammar and spelling mistakes such as P7 Line 143 "developmen"

Answer: Yes, thank you for pointing out these mistakes. I will check for similar problems.

---

## Author Comment (AC2) · 1 Feb 2021

General comments:

(1) The manuscript "Rainfall-runoff processes in the Loess Plateau, China: Temporal dynamics of event rainfall–runoff characteristics and diagnostic analysis of runoff generation patterns" reported a study concerning the mechanism changing of overland flow generation in Loess Plateau catchments. The method in which the authors identified dominant runoff generation process in the Loess Plateau catchments is confused

and not reliable. There is a lack of quantitative approach of identifying the mechanism changing of runoff generation and this made the reliability of the conclusion unsatisfactory.

Answer: Thank you for your comments. We have taken the time to think through all of your comments and carefully revised the manuscript as you suggested. Thank you for your valuable suggestion to improve the quality of the manuscript. The method used for identification dominant runoff generation process would be clearly rewritten. The quantitative approach would be considered for supporting conclusions in next version.

(2) On the other hand, the characteristics of rainfall–runoff processes are highly related to overland flow movement and concentration in catchments, particularly at the small and medium spatial scales. How did the authors deal with this? How to separate the effect of runoff-generation mechanism changing and the effect of overland flow movement condition changing led by LUCC?

Answer: The volume runoff was decided by runoff generation process and the time of runoff to the catchment outlet was decided by the runoff concentration process. So we proposed the method to identify dominant runoff generation mechanism considering of the combination of flow movement and concentration in the catchment. Maybe this method was not clearly enough to understand. Thank you for suggestions to improve our manuscript's quantity. We would clearly describe our novelty method in the next version.

Specific points:

(1) Line 5-10: are the mechanisms of runoff generation in these Loess Plateau catchments really changed? Any field evidence or reference provided in the main text?

Answer: Yes, there were many studies using field experiment measures to reveal the mechanisms of runoff in the Loess Plateau. However, these field-scale may be not enough to identify the catchment-scale runoff generation characteristics. We proposed

a new method for identification of dominant runoff generation mechanism. We could add some references to support our method in next version.

(2) Check the citation format of references in introduction. For example in line 22, the format of '(Jiongxin, 2005)' is wrong.

Answer: Yes, thank you for pointing out this mistake. I will check for the similar mistakes.

(3) Line 98: why did the authors choose these spatial scales (100âĹij10000 km2) and why these catchments? Please give the reason.

Answer: Yes, the representative analysis of the five selected catchments would be added in next version. These five basins have been used in many studies and they are special for the Yellow River Basin to understand rainfall-runoff changing.

(4) Line 110: why did the authors employ this approach to estimate antecedent soil moisture for rainfall-runoff events? Is there any reference?

Answer: Yes, the method used in this study was similar with the API model. Because of long-term period, observation of soil moisture was absent. Comparing with various methods of estimating soil moisture, the API method was sample and suitable for reflecting catchment scale moisture.

(5) Line 116: why there three steps? Please give references to make it clear.

Answer: The three steps were proposed by this study. However, this was not clear enough and I would rewrite the description of method in next version.

(6) Section 3.2: it seemed that this section introduced a method of qualitatively identifying saturation excess overland flow. However, there is no reference concerning this method. Is there any verification of this method? Did the authors have any field study to check the result of this method?

Answer: Yes, there was not reference concerning this method because this method
was first applied for identifying dominant runoff generation mechanisms by this study. This proposed study method was based on catchment hydrology mechanisms. There were some literatures that studied runoff generation at field-scale. The results maybe verify my method. I would make a supplement of this verification process in the next version.

(7) Section 3.2: using the method above, how did the authors identify the combination of infiltration-excess overland flow and saturation-excess overland flow in catchments? It looks like the authors used the 'probability' of saturation-excess overland flow introduced in the method to represent the combination of the two types of overland flow. Is this verified and reliable?

Answer: Yes, we used the 'probability' to infer the runoff generation mechanisms because of complicate rainfall-runoff process. This question was similar with question (7) and we would separately make a supplement of the verification process in the next version.

(8) Line 207: is there any research or measured data to support this point?

Answer: Yes, there were some studies that could support this conclusion. We would make a clear explanation to this point.

(9) Line 217-220: why you choose these rainfall runoff events.

Answer: Considering of collected data, we made a standard to select rainfall-runoff events. we would clearly classify this standard in next version.

(10) Check dams have been widely and massively constructed in the Loess Plateau, modifying rainfall-runoff processes in catchments (e.g., the Chabagou catchment in this study). How did the authors isolate their impacts on rainfall-runoff processes from LUCC impacts?

Answer: Yes, check dams and other anthropogenic engineering measures all significantly influenced the rainfall-runoff process. Actually, these human activities eventually

changed the catchment LUCC. So we did not isolate check dam's effects from LUCC impacts. We just studied the long-term variability of catchment-scale runoff generation.